# Mice with hyper-long telomeres show less metabolic aging and longer lifespans

Miguel A. Muñoz-Lorente[1], Alba C. Cano-Martin[1] & Maria A. Blasco[1]*

Short telomeres trigger age-related pathologies and shorter lifespans in mice and humans. In the past, we generated mouse embryonic (ES) cells with longer telomeres than normal (hyper-long telomeres) in the absence of genetic manipulations, which contributed to all mouse tissues. To address whether hyper-long telomeres have deleterious effects, we generated mice in which 100% of their cells are derived from hyper-long telomere ES cells. We observe that these mice have longer telomeres and less DNA damage with aging. Hyper-long telomere mice are lean and show low cholesterol and LDL levels, as well as improved glucose and insulin tolerance. Hyper-long telomere mice also have less incidence of cancer and an increased longevity. These findings demonstrate that longer telomeres than normal in a given species are not deleterious but instead, show beneficial effects.

[1] Telomeres and Telomerase Group, Molecular Oncology Program, Spanish National Cancer Centre (CNIO), Melchor Fernández Almagro 3, Madrid 28029, Spain. *email: mblasco@cnio.es

Telomeres are nucleoprotein structures at the ends of eukaryotic chromosomes[1,2]. They consist of tandem repeats of the TTAGGG DNA sequence bound by six-protein complex known as shelterin[2,3]. Telomeres are essential to protect chromosome ends from DNA degradation and DNA repair activities and they play an important role in chromosome stability[4,5]. Telomeres shorten with every cell division owing to the so-called "end replication problem"[6–8]. Telomerase is a ribonucleoprotein DNA polymerase that can elongate telomeres by de novo addition of TTAGGG repeats onto the chromosome ends[9], thus compensating for telomere attrition. Telomerase consists of a reverse transcriptase catalytic subunit (known as TERT) and an associated RNA component (Terc), which is used as a template for the synthesis of TTAGGG repeats. In the adult organism, telomerase activity is restricted to stem cell compartments although this is not sufficient to prevent progressive telomere shortening with aging[10,11]. When telomeres reach a critically short length, they induce a persistent DNA damage response (DDR) which induces other cellular events, such as cellular senescence and/or apoptosis, as well as impairs the ability of stem cells to regenerate tissues[5,12–14]. Telomere shortening is considered one of the hallmarks of aging as short telomeres are sufficient to cause organismal aging and decreased lifespan[15–18]. Telomere length is determined genetically and both average telomere length and the rate of telomere shortening varies between species[19,20]. In this regard, humans are born with shorter telomeres than mice, but mice telomeres shorten 100-times faster than humans[21–23].

In mice, we and others have previously demonstrated that telomerase is activated at the blastocyst stage where telomeres are elongated in the inner cell mass (ICM), in this manner setting the normal telomere length of a given species[24–26]. Interestingly, we also found that in vitro expansion of embryonic stem (ES) cells derived from the ICM allows for additional telomere elongation beyond the normal telomere length of the species, generating the so-called hyper-long telomere ES cells[25,26]. This telomere elongation is associated with epigenetic changes during the pluripotency stage which allow for a more "open" telomeric chromatin structure and for telomerase-mediated telomere extension in the absence of conspicuous gene expression changes (ref. [11]; ref. [25,26]). Recently, we demonstrated that these ES cells with hyper-long telomeres can be aggregated into morulae to obtain chimeric mice which are also made of cells with longer telomeres than those of the natural species[26]. Thus, ES cells with hyper-long telomeres are able to contribute to all adult organs, and do not affect their normal architecture and function[26]. However, further studies are necessary to address the long-term effects of increasing the natural telomere length of a species.

To this end, here we generate chimeric mice with a 100% contribution from ES cells with hyper-long telomeres, that have longer telomeres than normal in the whole organism (referred here as "hyper-long telomere mice"). We observe that hyper-long telomere mice are significantly leaner than control mice. We further show that this reduction in body size is concomitant with a lower accumulation of white adipose tissue (WAT) in the organism. Interestingly, mice with hyper-long telomeres also show an improved glucose metabolism compared to control mice, as indicated by an improved glucose and insulin tolerance throughout life. In agreement with less metabolic aging, these mice show longer telomeres in the liver and the white adipose tissue even at older ages. Importantly, mice with hyper-long telomeres show an increased longevity and develop less tumors associated with aging. Together, these findings demonstrate that longer telomeres than normal show beneficial effects in mice, delaying metabolic aging and cancer, and resulting in longer lifespans.

## Results

**Mice with longer telomeres than those of the natural species**. We previously showed that it is possible to generate mouse ES cells with hyper-long telomeres in the absence of any genetic manipulations, and that these ES cells can be used to generate mouse chimeras, which carry cells with longer telomeres than those of the natural species[26]. In order to address the impact of excessively long telomeres in a given species, here, we have generated a cohort of chimeric mice with a 100% contribution from ES cells with hyper-long telomeres ES cells. To this end, ~100 morulae at the eight-cell state were microinjected with 6–10 GFP-positive female ES cells with hyper-long telomeres at passage 16 by laser-assisted perforation of the "zona pellucida" obtaining chimeric mice which were 100% contributed by the GFP-positive ES cells (Fig. 1a). In particular, adult mice (1.5–2 years of age) showed 100% of their cells positive for eGFP as determined by immunohistochemistry (IHC) with anti GFP antibodies in different mouse tissues (Fig. 1b). The fact that, even at old age, chimeric mice show 100% of the cells that are GFP-positive, and therefore are derived from ES cells with hyper-long telomeres, indicates that these cells with hyper-long telomeres are not negatively selected during aging. Further supporting that cells with hyper-long telomeres are not deleterious, all mouse tissues showed a normal histology even at older ages (Fig. 1b). Strikingly, hyper-long telomere mice were significantly leaner than control mice from the same genetic background (Fig. 1c). To further study this unexpected phenotype, we carried out a longitudinal follow-up of mouse weight throughout their entire lifespan and found that hyper-long telomere mice showed a significant reduction in body weight which started from 40 weeks of age onwards (Fig. 1d).

Thus, here we generated viable mice that are 100% contributed by ES cells with hyper-long telomeres. We did not find any overt phenotypes in these mice other than they are leaner compared to normal mice.

**Less fat accumulation in hyper-long telomere mice**. In order to investigate the leaner phenotype of hyper-long telomere mice, we performed densitometry assays at 75 and 100 weeks of age in both hyper-long telomere mice and normal telomere length controls from the same genetic background. We found that hyper-long telomere mice showed significantly lower fat content than age-matched control mice at two different ages (75 and 100 weeks-old) (Fig. 2a). In contrast, no differences were observed in either bone mineral density (BMD) (Fig. 2b) or in total lean mass between hyper-long telomere mice and the normal telomere length controls (Fig. 2c). In agreement with lower fat accumulation, hyper-long telomere mice also showed a significantly reduced skin subcutaneous fat layer compared to age-matched control mice (100 weeks of age) (Fig. 2d, e). These results indicate that the reduced body size of hyper-long telomere mice is due to a lower accumulation of fat.

**Improved metabolic parameters in hyper-long telomere mice**. As mice with hyper-long telomeres showed reduced fat accumulation, we next set to determine different metabolic parameters. First, we measured the levels of albumin, creatinine, bilirubin, urea, alanine aminotransferase (ALT), cholesterol, the low-density lipoprotein (LDL), and the high-density lipoprotein (HDL) in blood serum from both hyper-long telomere mice and age-matched controls at 50, 75, and 100 weeks of age. We found no significant differences in the levels of creatinine, bilirubin, albumin, urea and, HDL at any of the time points between hyper-long telomere mice and control mice (Fig. 3a–e). Interestingly, hyper-long telomere mice showed significantly lower levels of the

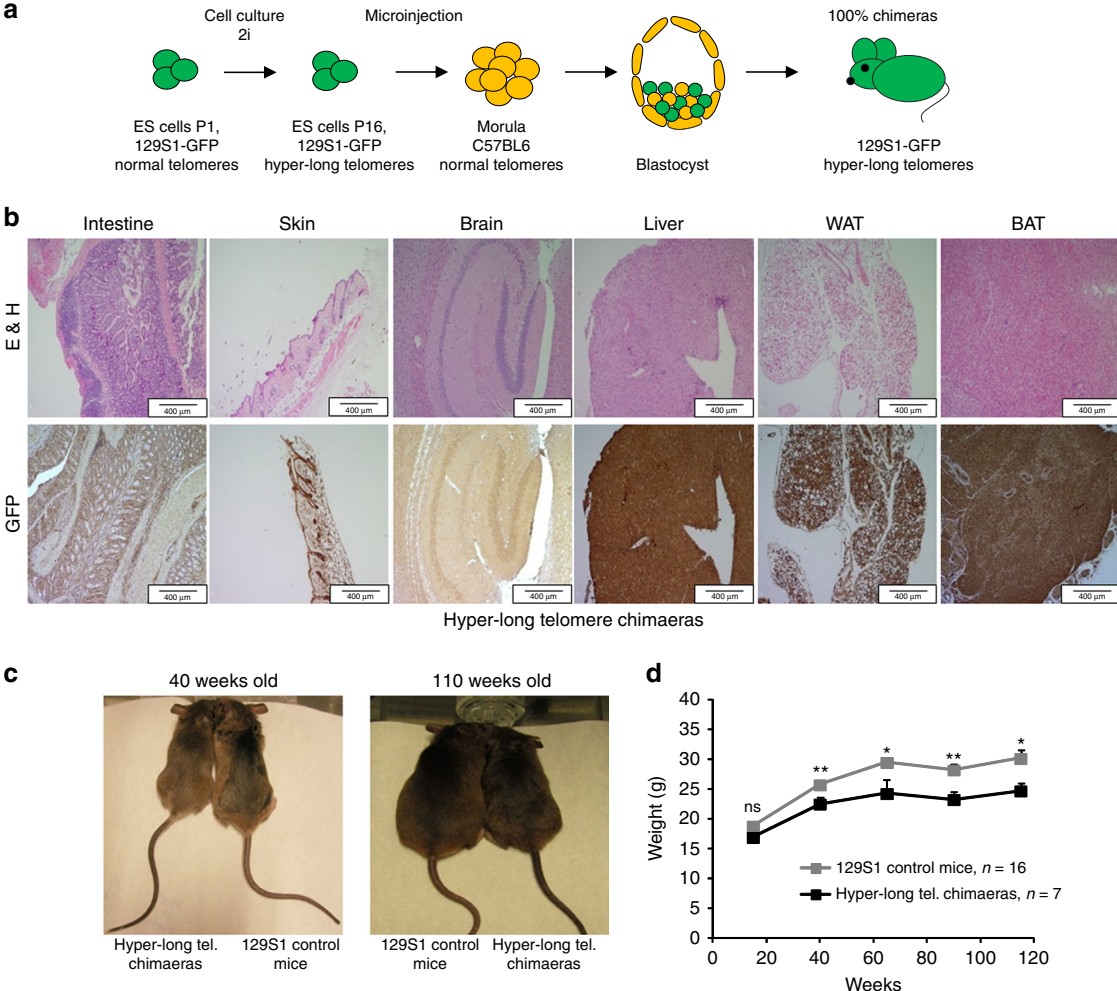

**Fig. 1** Hyper-long telomere mice are leaner than normal mice. **a** Scheme showing generation of mice with hyper-long telomeres. eGFP ES cells are cultured in 2i medium until passage 16 and they are microinjected into morulae to obtain 100% contributing chimaeras. **b** Representative images showing 100% eGFP ES cell contribution in different organs and tissues. **c**, **d** Longitudinal follow up of weight in the indicated cohorts. Hyper-long telomere mice show a significant reduction in body weight from 40 weeks onwards (**c**) and this body size reduction is maintained until the last timepoint measured (**c**, **d**). Error bars represent standard error. *t*-test was used for statistical analysis. The number of mice is indicated in each case. *$p < 0.05$. **$p < 0.01$. Source data are provided as a Source Data file

"bad cholesterol", LDL, compared to control mice at the three time points studied (Fig. 3f). Hyper-long telomere mice also showed significantly lower levels of cholesterol and of the hepatic damage marker ALT compared with control mice (Fig. 3g, h). Thus, we found lower cholesterol levels, including LDL levels, and decreased liver damage in hyper-long telomere mice compared to control mice.

To further asses the metabolic effects of hyper-long telomeres, we next studied glucose metabolism by performing a glucose tolerance test (GTT). To this end, glucose was injected intraperitoneally in age-matched hyper-long telomere mice and normal telomere mice at 50, 75, and 100 weeks of age previously fasted for 16 h. At all timepoints, hyper-long telomere mice showed an increased glucose sensitivity (Fig. 3i–k). We obtained similar results when we performed an insulin tolerance test (ITT). In this case, hyper-long telomere mice showed a better insulin tolerance at both 75 and 100 weeks of age compared to the normal telomere length controls (Fig. 3l–n).

In summary, these results indicate that hyper-long telomere mice have an improved metabolic fitness compared to normal telomere length mice. In particular, they show lower levels of LDL, cholesterol, and ALT in blood, as well as are more sensitive

to glucose and insulin even at old age (100-weeks old), suggesting less metabolic aging compared to age-matched control mice.

**Hyper-long telomere mice live longer and have less spontaneous tumour incidence.** Critically short telomeres in humans and mice can lead to premature aging and shorter lifespans by limiting the ability of stem cells to regenerate tissues[11,12,27]. In turn, we previously showed that longer telomeres owing to telomerase over-expression in adult mice can increase mouse longevity[17,28]. Thus, here we set to address whether longer telomeres than normal in the adult organism, in the absence of telomerase over-expression, were sufficient to increase longevity. Strikingly, we found that hyper-long telomere mice showed a significant increase of 12.75% in median longevity as well as an increase in maximum longevity of 8.40% compared to normal telomere length controls (Fig. 4a). These findings indicate that long telomeres *per se*, even in the absence of telomerase over-expression, are sufficient to increase mouse longevity.

The fact that hyper-long telomere mice had an increased longevity also suggested that they were unlikely to have any deleterious effects promoting cancer. This question is of

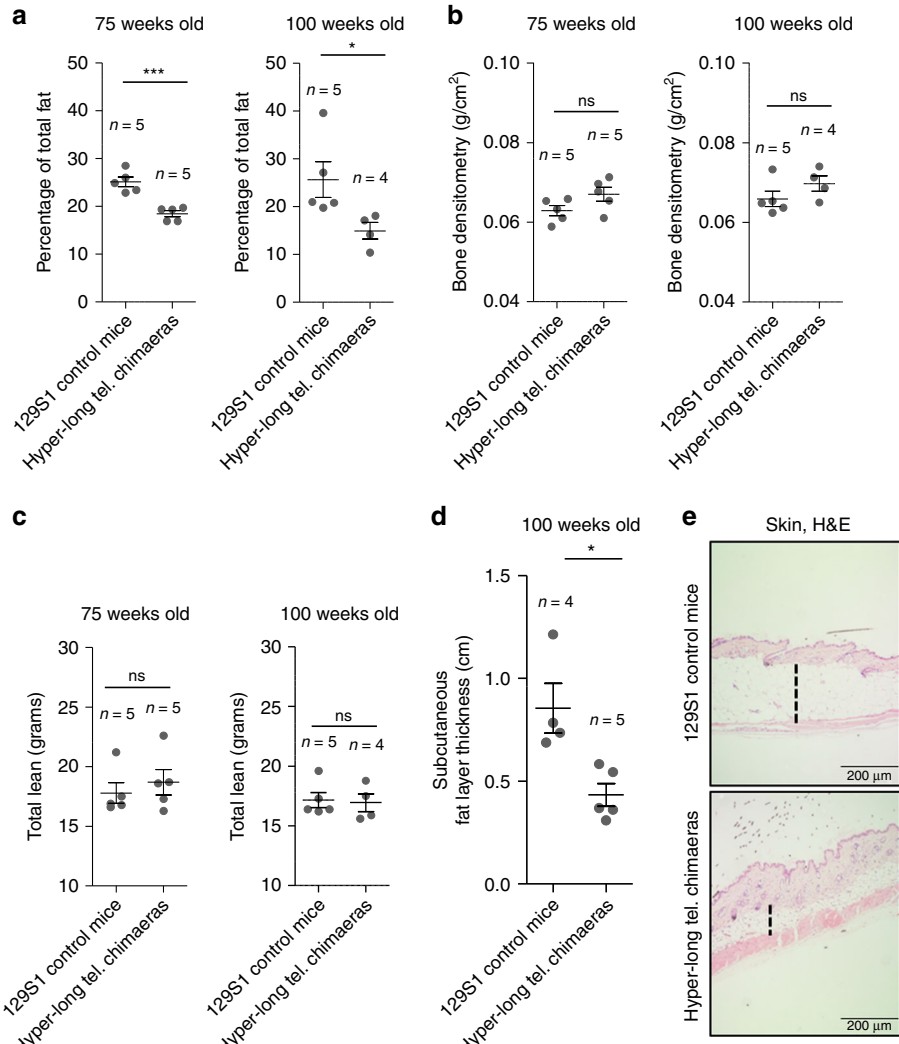

**Fig. 2** Reduced body size in hyper-long telomere mice is due to lower fat accumulation. **a–c** Densitometry analysis of hyper-long telomere mice and normal mice at two different ages. **a** Quantification of percentage of total fat. **b** Quantification of bone density. **c** Quantification of total lean mass. **d**, **e** Quantification of the skin subcutaneous fat layer in 100 weeks-old hyper-long mice and age-matched controls (**d**), and representative image showing skin subcutaneous fat layer thickness. Measurements were done by ImageJ software and calculated by the mean value of 30–40 different measurements all over the skin subcutaneous fat layer. Error bars represent standard error. t-test was used for statistical analysis. The number of mice is indicated in each case. *p < 0.05. ***p < 0.001. Source data are provided as a Source Data file

particular importance because previous works have correlated presence of long telomeres in humans with increased cancer incidence[29–33]. To address this, we studied the spontaneous tumor incidence of hyper-long telomere mice and control mice. We found that hyper-long telomere mice showed a reduction of almost 50% in the number of mice that developed tumors compared to the normal telomere length control mice, although this difference did not reach significance (Fig. 4b). Of notice, the cause of death of tumor free mice was associated to general body degeneration associated to the aging process, as well as uterine infection in some female mice.

These findings clearly demonstrate that long telomeres per se do not lead to increased cancer, at least in mice. Instead, longer telomeres are clearly correlated with an increased mouse longevity.

**Hyper-long telomere mice have normal cognitive capabilities.** Next, we wondered whether having longer telomeres than normal could affect cognitive capabilities in these mice. To address this, we performed different cognitive tests at 50, 75,

and 100 weeks of age. To evaluate neuromuscular endurance, we performed a rotarod test, which measures the time that mice are able to stay on a rotating platform with accelerated movement without falling (Supplementary Fig. 1a). We observed no differences between the hyper-long telomere mice and the control mice at the different time points (Supplementary Fig. 1b). In order to study mouse coordination, we performed a tightrope test, which evaluates the capability of the mice to stay on a rope without falling during at least 1 min (Supplementary Fig. 1c). Again, we did not see any significance difference in performance between the hyper-long telomere mice and the controls (Supplementary Fig. 1d). Finally, to measure the sensory perception of mice we used a buried food test, in which we measured the ability of mice to find a buried food pellet after 16 h fasting (Supplementary Fig. 1e). Again, we found no differences between hyper-long telomere mice and controls (Supplementary Fig. 1f).

Together, these results suggest that hyper-long telomere mice show normal cognitive capabilities, such as normal coordination, balance and smell.

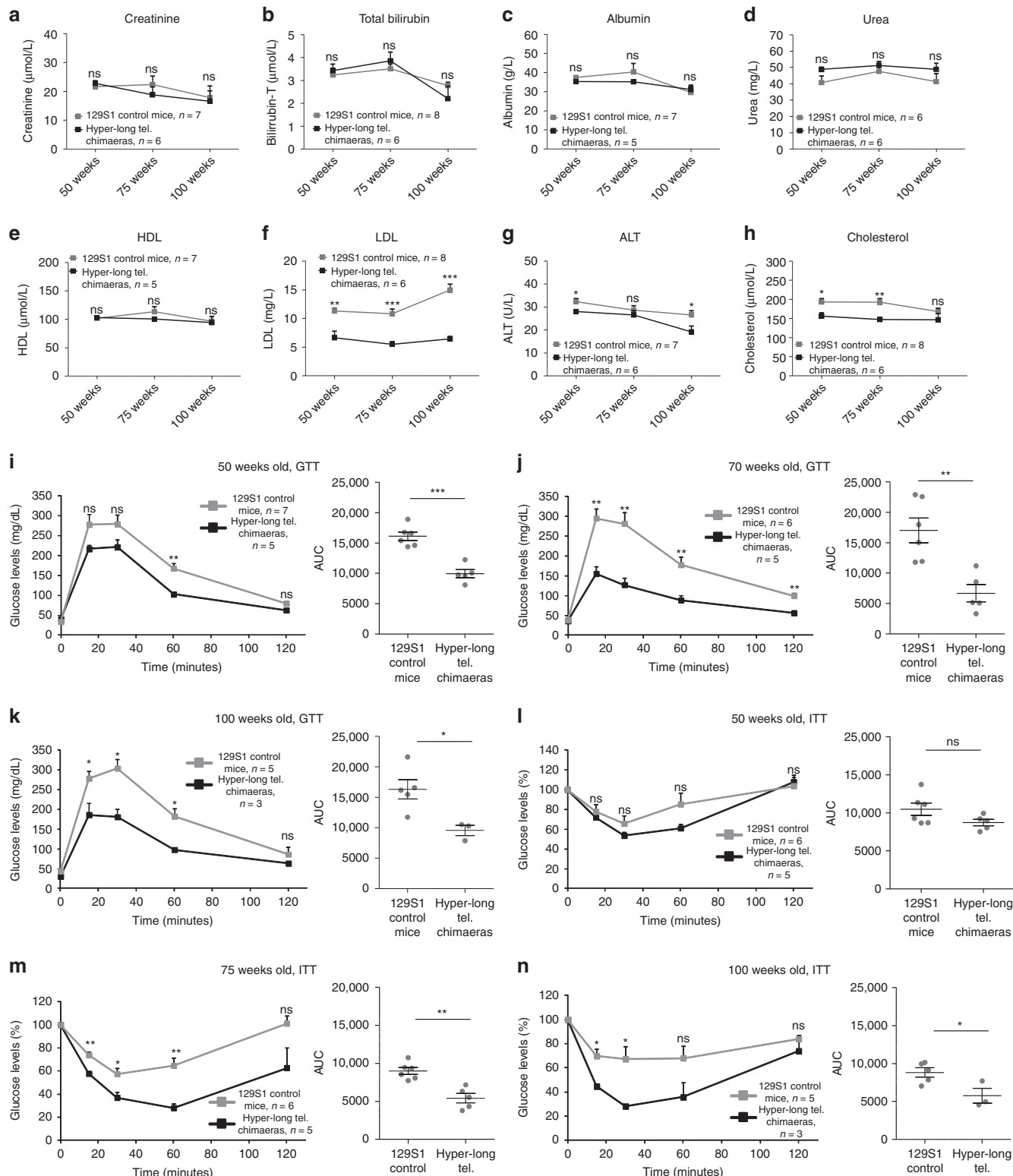

**Fig. 3** Hyper-long telomere mice show less metabolic aging. **a–h** Pentra quantification of different serum metabolites at 50, 75, and 100 weeks-old. Metabolites were measured in serum obtained from blood extracts using ABX Pentra, **a** quantification of creatinine levels, **b** quantification of total bilirubin levels, **c** quantification of albumin levels, **d** quantification of urea levels, **e** quantification of HDL (high density lipoprotein) levels, **f** quantification of LDL (low density lipoprotein) levels, **g** quantification of ALT (alanine aminotransferase) levels and **h** quantification of cholesterol levels. **i–k** Glucose tolerance test (GTT). GTT was performed in hyper-long telomere mice and age-matched controls at 50 (**i**), 75 (**j**) and 100 (**k**) weeks of age by intraperitoneal glucose injection after 16 h fasting. **l–n** Insulin tolerance test (ITT). ITT was performed in hyper-long telomere mice and age-matched controls at 50 (**l**), 75 (**m**) and 100 (**n**) weeks of age by intraperitoneal insulin injection. Error bars represent standard error. $t$-test was used for statistical analysis. The number of mice is indicated in each case. *$p < 0.05$. **$p < 0.01$. ***$p < 0.001$. Source data are provided as a Source Data file

**a**

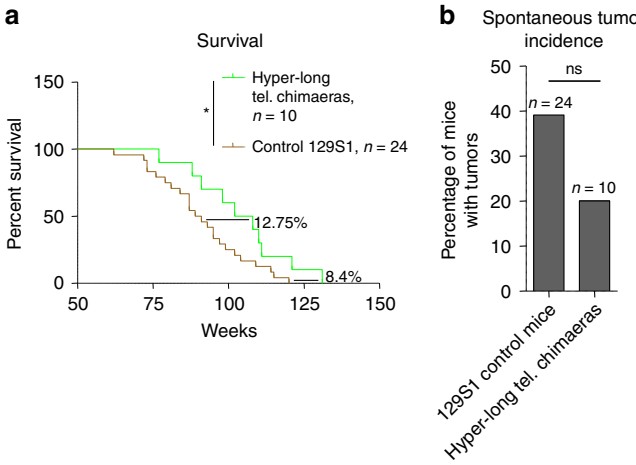

**b**

Fig. 4 Hyper-long telomere mice live longer and have less spontaneous cancer. **a** Survival proportions of 100% hyper-long chimeric mice compared to control mice from the same background. Hyper-long telomere mice show a 12.75% increase in median lifespan and an 8.4% increase in maximum lifespan. **b** Quantification of spontaneous tumor incidence in hyper-long telomere mice and controls. Hyper-long telomere mice show a decrease in almost 20% in tumor incidence compared to control mice from the same background. Error bars represent standard error. Mantel–Cox test was used for statistical analysis in survival curves and Chi-square test for spontaneous tumor incidence. The number of mice is indicated in each case. *$p < 0.05$. **$p < 0.01$. ***$p < 0.001$. Source data are provided as a Source Data file

**Hyper-long telomere mice retain longer telomeres at old ages**. In order to determine whether hyper-long telomere mice retained longer telomeres even at old age, we measured telomere length in different tissues at 100 weeks of age. Telomere length was determined in intestine and skin, as examples of proliferative tissues, and in brain as a non-proliferative tissue. We found that in all cases, 100-weeks old hyper-long telomere mice showed longer telomeres on average, than the normal age-matched controls (Fig. 5a–f). In agreement with longer telomeres, hyper-long telomere mice also showed a significantly lower accumulation of short telomeres at old age compared with the age-matched control mice (Fig. 5a–f).

Importantly, as one of the main phenotypes in the hyper-long telomere mice is an improved metabolic profile, we measured telomere length and percentage of short telomeres in metabolically relevant tissues, such as the liver, white adipose tissue (WAT), and brown adipose tissue (BAT) at 100–110 weeks old. Interestingly, we found a very pronounced increase in telomere length in the hyper-long telomere mice in all three metabolic tissues compared to control mice from the same background (Fig. 6a–f). Accordingly, we observed a significant decrease in the percentage of short telomeres in the hyper-long telomere mice compared to controls in all three tissues (Fig. 6a–f).

To rule out that longer telomeres in the hyper-long telomere mice where the result of an altered telomerase expression in the adult organism we determined mRNA levels of the two essential telomerase components *Tert* and *Terc* in hyper-long telomeres and normal telomere mice at 100–110 weeks of age in the liver and the WAT. As expected, we found that *Tert* was not expressed in these tissues in agreement with previous reports showing that *Tert* expression is undetectable in the majority of adult mouse tissues after birth[10,17]. Furthermore, we did not see any significant differences in the mRNA expression of *Terc* between the hyper-long telomere mice and the normal controls in the liver and the WAT (Fig. 7a–d).

Altogether, these results demonstrate that hyper-long telomere mice retain longer telomeres with aging, including metabolic tissues such liver, white adipose tissue and brown adipose tissue, in the absence of *Tert* overexpression.

**Hyper-long telomere mice show less senescence and DNA damage**. Next, we addressed whether longer telomeres than normal could be promoting or protecting from age-associated DNA damage. To this end, we performed a telomere Q-FISH to identify telomeres followed by an immunofluorescence against the DNA damage marker 53BP1 in liver of 100 weeks hyper-long telomere chimaeras and controls. To this end, we quantified the number of cells presenting ≥2 53BP1 foci (Fig. 8a, c). Interestingly, hyper-long telomere mice show an 8-fold decrease in 53BP1-positive cells compared to controls (Fig. 8a). Importantly, the percentage of cells presenting ≥1 telomere induced foci (TIF) also show a very significant 6-fold decrease in hyper-long telomere mice compared to controls (Fig. 8b, c). To further investigate whether longer telomeres than normal protect from DNA damage, we quantified the percentage of cells positive for the senescence marker, p21, in the liver of age matched (100 weeks of age) normal length and hyper-long telomere mice. Interestingly, we found a significant 9-fold decrease in the number of p21 positive cells in hyper-long mice compared with age-matched controls (Fig. 8d, e).

Together, these results indicate that longer telomeres than normal in hyper-long telomere mice significantly reduce the global DNA damage and the telomeric DNA damage associated with aging in mice.

**Enhanced mitochondrial function in hyper-long telomere mice**. Since previous works have reported a relation between telomere and mitochondrial homeostasis[34,35], here we next addressed whether hyper-long telomeres could be also affecting mitochondrial function. To this end, we first performed a qPCR-based assay in order to determine the relative mitochondrial DNA (mtDNA) copy number in WAT of 100–110-week old mice using three different mitochondrial genes (*Cox1, Cytb,* and *Nd1*). Interestingly, we find that relative mtDNA copy number in WAT of hyper-long telomere mice is double those of control mice from the same background according to the three genes mitochondrial analyzed (Fig. 9a–c).

Next, we also measured mRNA expression levels of the different oxidative phosphorylation genes (OXPHOS) Cytochrome C, ATP synthase, Cytochrome C subunit 6 and Cytochrome C subunit 5a as well as mitochondrial regulators PGC-1α and PGC-1β and critical targets such as ERRα and PPARα. Concomitant with our previous result, we found a significant upregulation in all these genes in hyper-long telomere mice compared to the normal length age-matched controls (Fig. 9d–k).

Taken together, these data suggest that mice with hyper-long telomeres have an improved mitochondrial function, which could be also contributing to their delayed aging phenotype and improved metabolic function.

**Normal expression of RAP1 in mice with hyper-long telomeres**. We previously showed that mice deficient for the RAP1, a component of the shelterin telomere protective complex, showed an obese phenotype as well as signs of metabolic syndrome including abnormal fat accumulation, glucose intolerance and fatty liver[36]. As we found here that hyper-long telomere mice are protected from age-associated metabolic syndrome, including fat accumulation, high cholesterol levels, liver damage and glucose intolerance, we set to address whether hyper-long telomeres were affecting the levels of

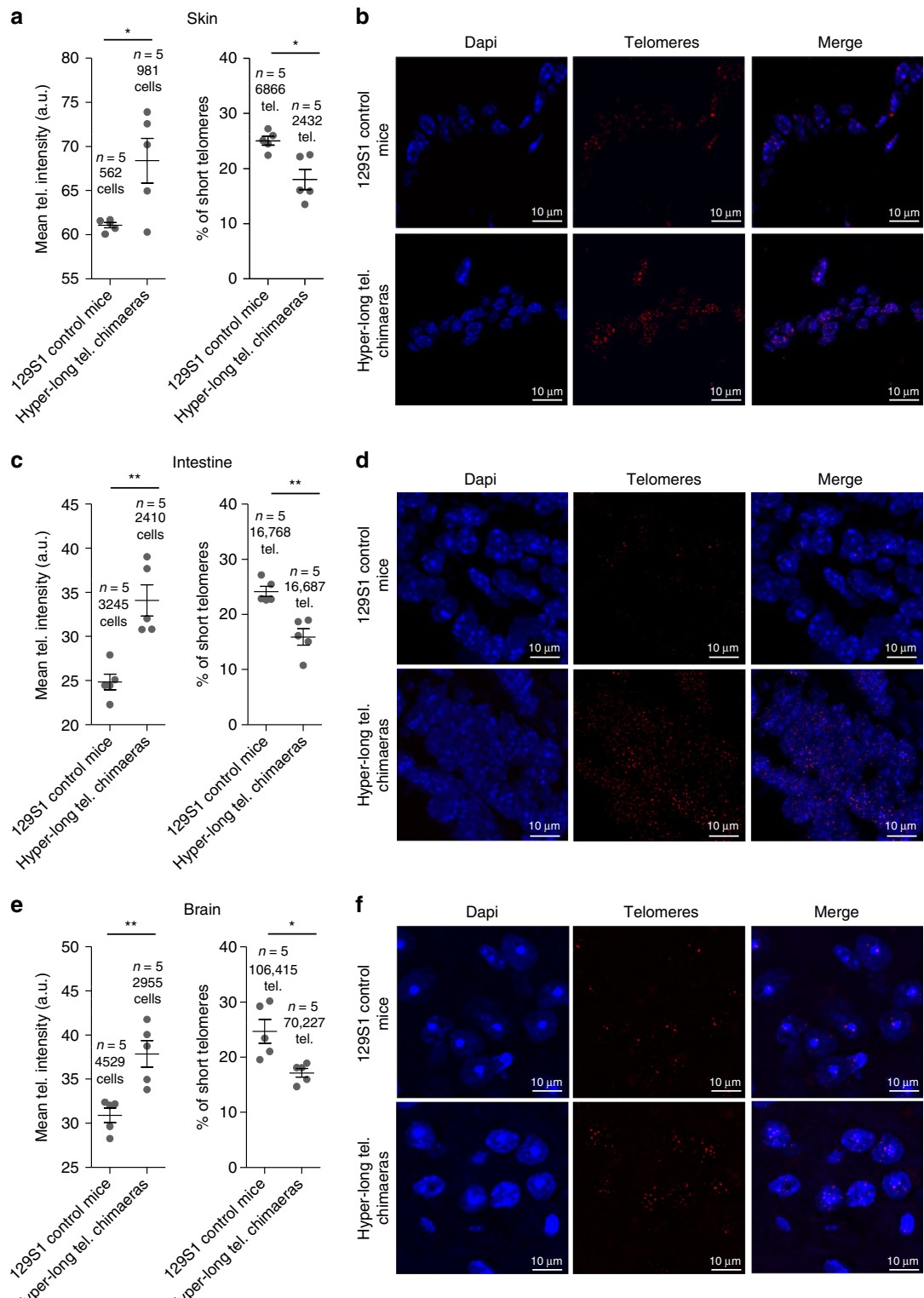

**Fig. 5** Hyper-long telomere mice show longer telomeres throughout their lifespan in both proliferative and non-proliferative tissues. **a–f** Mean telomere fluorescence and percentage of short telomeres in back skin (**a**, **b**), intestine (**c**, **d**), brain (**e**, **f**) in 100 weeks-old hyper-long telomere mice and age-matched controls. Error bars represent standard error. *t*-test was used for statistical analysis. The number of mice is indicated in each case. *$p < 0.05$. **$p < 0.01$. ***$p < 0.001$. Source data are provided as a Source Data file

the RAP1 protein. To this end, we first determined RAP1 protein levels by Western blot in the white adipose tissue (WAT). We found that RAP1 protein levels were similar in age-matched 100–110 weeks-old hyper-long telomere and normal telomere

length controls (Supplementary Fig. 2a, b). Similarly, we detected similar RAP1 protein levels in the liver of hyper-long and normal telomere length mice as determined by immunofluorescence (IF) analysis at 100–110 weeks (Supplementary Fig. 2c, d). These

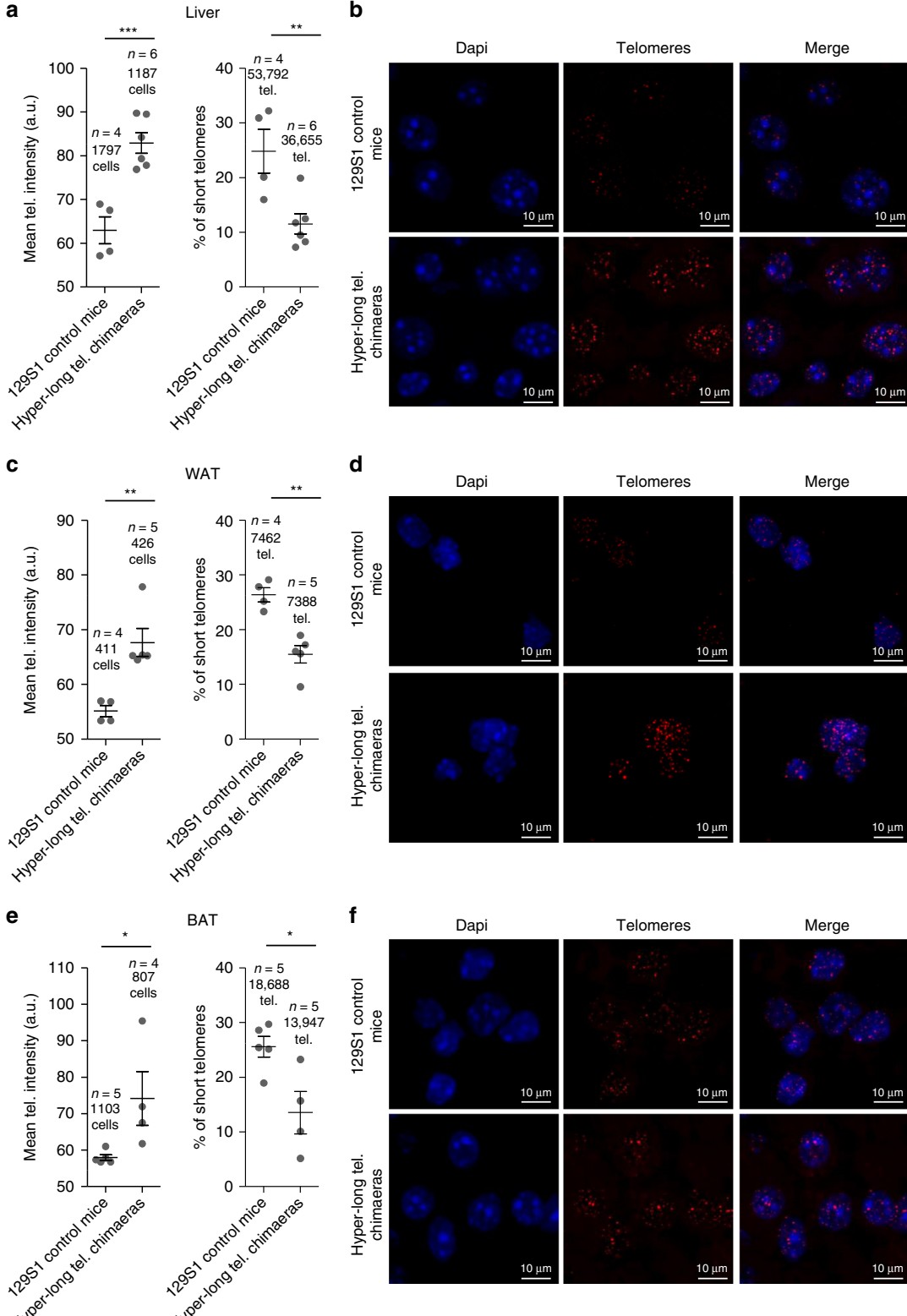

**Fig. 6** Hyper-long telomere mice show longer telomeres throughout their lifespan in metabolic-related tissues. **a–f** Mean telomere fluorescence and percentage of short telomeres in liver (**a**, **b**), white adipose tissue (**c**, **d**) and brown adipose tissue (**e**, **f**), in 100 weeks-old hyper-long telomere mice and age-matched controls. Error bars represent standard error. *t*-test was used for statistical analysis. The number of mice is indicated in each case. *$p < 0.05$. **$p < 0,01$. ***$p < 0,001$. Source data are provided as a Source Data file

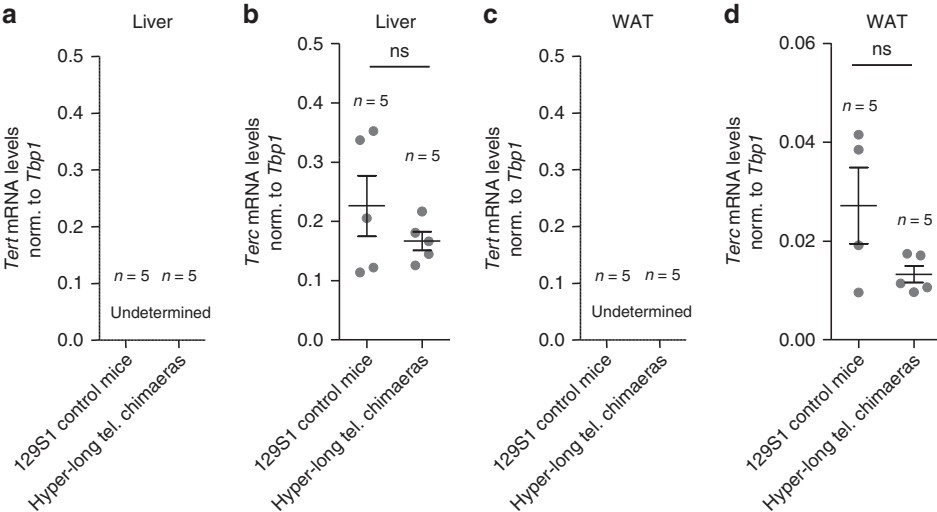

**Fig. 7** Hyper-long telomere mice does not present altered telomerase expression levels. **a–d** *Tert* and *Terc* mRNA expression levels as measured by Q-PCR in liver (**a**, **b**) and in white adipose tissue (**c**, **d**) in age-matched (100 weeks old) hyper-long telomere mice and control mice. Error bars represent standard error. *t*-test was used for statistical analysis. The number of mice is indicated in each case. $*p < 0.05$. $**p < 0,01$. $***p < 0,001$. Source data are provided as a Source Data file

findings suggest that protection from metabolic syndrome in mice with hyper-long telomeres does not seem to be related to altered levels of the RAP1 protein.

Next, we addressed whether hyper-long telomeres resulted in altered levels of the rest of the components of shelterin. To this end, we determined the mRNA expression levels of the different shelterin components in the liver and WAT of 100–110-week old mice. Again, although we observed a tendency to lower expression levels of the different shelterin components in hyper-long telomere mice compared to the controls both in the liver, this difference did not reach statistical significance (Supplementary Fig. 2e–k), similar findings were observed and in the white adipose tissue (Supplementary Fig. 2l–r). Thus, hyper-long telomeres in mice do not alter the mRNA expression of *Rap1* or the different shelterin components.

## Discussion

It has been previously described that telomere elongation is only possible in certain adult stem cell compartment in which telomerase is partially active in both human and mouse and even in these cells telomerase expression is not enough to maintain telomere homeostasis throughout the lifespan of organisms[10,17,21,23,37]. Indeed, although mice are born with longer telomeres than humans the rate of telomere shortening in mice is up to 100-folder higher[21,23] and both humans and mice show progressive telomere shortening with aging[17,23,38]. In turn, telomere shortening with aging can trigger a number of secondary pro-aging phenomena such as increased DNA damage and genomic instability, cellular senescence and/or apoptosis, impaired ability of stem cells to regenerate tissues etc., and therefore it is considered one of the primary hallmarks of organismal aging (ref. [18]). In turn, telomere maintenance in the adult organisms by using different telomerase over-expression approaches, including gene therapy strategies, have been shown to delay aging and age-associated pathologies, as well as to increase lifespan in mice[17,28,39–42].

However, telomerase has been found re-activated in the majority of human cancers[43] and this has generated a certain concern on potential negative long-term effects of telomerase reactivation in promoting tumorigenesis. In this regard, there is mounting evidence that telomerase re-activation in the adult organism by using non-integrative gene adeno associated vectors (AAV) does not lead to increased cancer, even in the context of activated oncogenes, in mice[44–47]. In humans, presence of longer telomeres than normal has been also associated to increased incidence of certain cancers such as lung cancer in large population studies[29–33]. In addition, germinal mutations in the *Pot1* shelterin gene, which lead to longer telomeres than normal, have been also associated to various types of familial cancer, such as melanoma, Li-fraumeni like, and glioma[48–54]. Although in the latter case scenario, Pot1 mutations not only lead to longer telomeres, but also increased telomere aberrations, which could be responsible of the increased cancer susceptibility[48,50–54]. Thus, it is of relevance to address whether long telomeres *per se*, in the absence of telomerase activation or other telomere alterations, could be promoting tumorigenesis or not.

In the past, we demonstrated that it is possible to elongate telomeres during in vitro expansion of ES cells beyond the normal length of the species in the absence of any genetic manipulations[25] and to use this ES cells with hyper-long telomeres to generate healthy chimeric mice derived from these ES cells with hyper-long telomeres[26]. Here, in order to address any potential long-term deleterious effects of longer telomeres than normal in the context of an organism, we generated chimeric mice that are 100% contributed from ES cells with hyper-long telomeres (hyper-long telomere mice), and followed these mice during their entire lifespan.

We found that these mice derived from ES cells with hyper-long telomeres, retained longer telomeres than normal at older ages and showed no pathological abnormalities. Accordingly, we also found a significantly decreased number of cells presenting global DNA damage, as well as telomere-induced DNA damage with aging in the hyper-long telomere mice compared to the normal telomere controls. Also, the levels of the p21 senescence marker were decreased in the hyper-long telomere mice.

Interestingly, hyper-long telomere mice showed a reduced body weight than normal from week 40 onwards. We further found that this reduced size is due to a lower accumulation of fat in the absence of changes in the lean mass. Furthermore, hyper-long telomere mice show signs of a "younger" metabolic phenotype, as indicated significantly reduced levels of LDL, ALT, and cholesterol in serum throughout their lifespan. Moreover, they show an increased sensitivity to glucose and insulin intakes, even

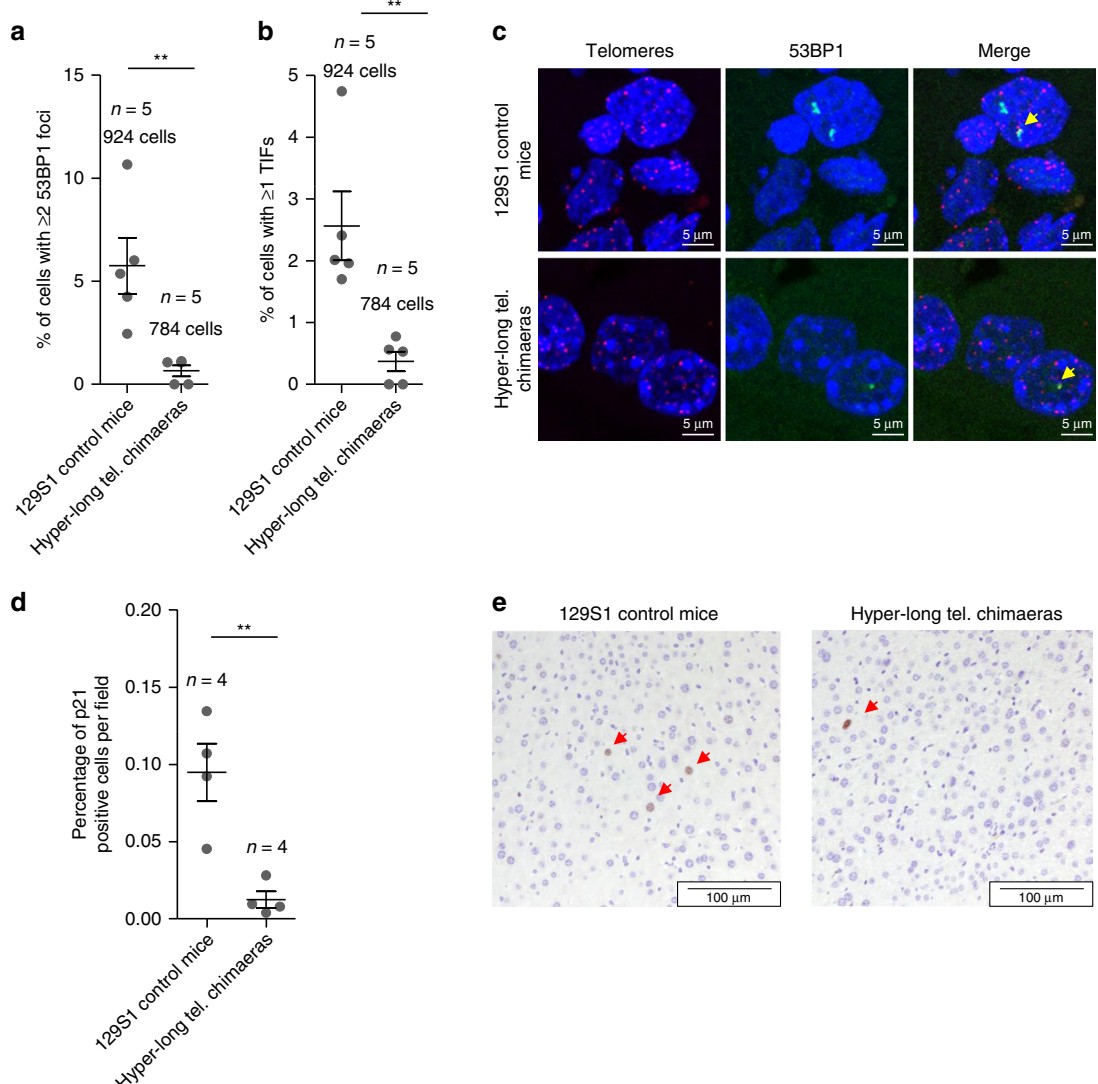

**Fig. 8** Hyper-long telomere mice show less DNA damage and less senescence. **a, b** Quantification of DNA damage in age-matched (100 weeks old) hyper-long telomere mice and control mice. Quantification of total damage as indicated by percentage of cells with ≥2 53BP1 foci as determined by immunofluorescence (**a**) and quantification of cells showing telomere-induced DNA damage as shown by percentage of cells with ≥1 TIF (telomere induced foci) as determined by telomere FISH followed by immunofluorescence with 53BP1 antibody (**b**). **c** Representative images of TIFs (yellow arrow). **d** Quantification of the percentage of p21 positive cells in liver of age matched (100 weeks old) hyper-long telomere mice and control mice. **e** Representative images of p21 (red arrows) positive cells. Error bars represent standard error. *t*-test was used for statistical analysis. The number of mice is indicated in each case. **$p < 0.01$. Source data are provided as a Source Data file

at old age, thus supporting a "younger" metabolic age in the case of hyper-long telomere mice compared to the normal telomere length controls.

Previous reports have shown decreased mitochondrial function associated to telomere dysfunction[34,35]. In agreement with these previous findings, our current data suggest that mice with hyper-long telomeres have an improved mitochondrial function. In particular, we found that hyper-long telomere mice show increased expression of *Pgc1-α/β* as well as of their target genes *Errα* and *Pparα*. Moreover, hyper-long telomere mice also show increased mitochondrial DNA copy number and increased expression levels of the OXPHOS mitochondrial genes Cytochrome C, ATP synthase, Cytochrome C subunit 6 and Cytochrome C subunit 5a compared to normal telomere controls. Together, these findings suggest that mitochondrial activity is enhanced in mice with hyper-long telomeres, and this could contribute to their improved metabolic performance.

Importantly, we did not see increased incidence of spontaneous tumors in the hyper-long telomere mice, which instead, showed a clear tendency to have reduced spontaneous tumors, compared to the normal telomere length control, thus indicating that long telomere per se do not increase tumorigenesis, instead, seem to be decreasing cancer risk in agreement with a younger state. Also in agreement with this, we found that hyper-long mice lived longer than controls with a 12.75% increase in median survival and 8.4% in maximum survival. It could be argued that the 100% chimeric mice used in this study maybe derived from a donor trophoblast whose gender could impact in the survival of these mice. However, it is known[55,56] that trophoblast cells contribute entirely to form the placenta and the amniotic sac and are not likely to affect adult mice.

Overall, here we present a mouse model with delayed aging in the absence of genetic manipulations. This mouse model features anti-aging phenotypes that are also present in several genetically

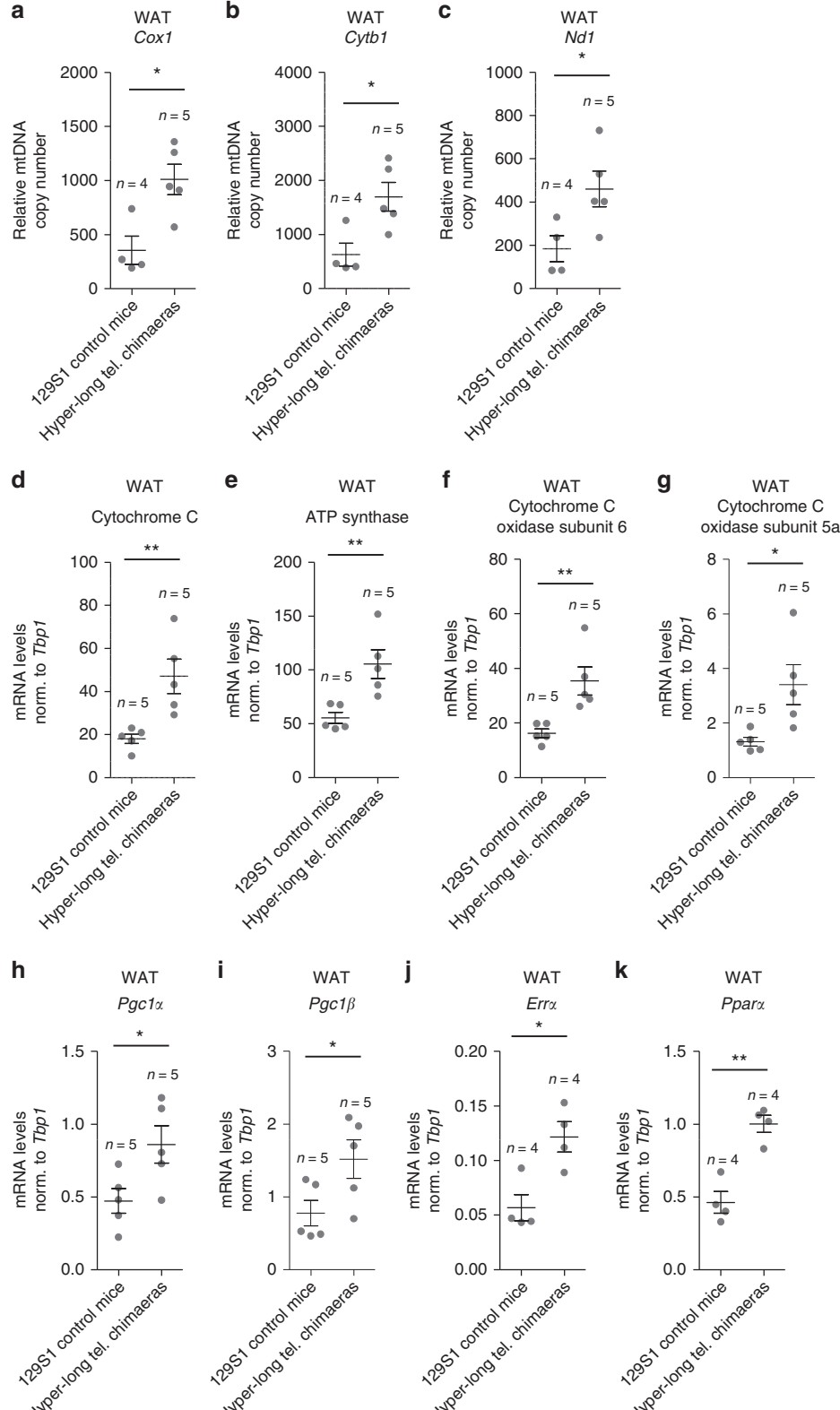

**Fig. 9** Improved mitochondrial function in mice with hyper-long telomeres. **a–c**. Relative mtDNA content was calculated by the comparative Ct method of the mitochondrial genes *Cox1* (**a**), *Cytb1* (**b**) and *Nd1* (**c**) compared to the nuclear gene *H19* in WAT of 100–110 weeks old hyper-long telomere mice and age-matched controls. **d–k**. mRNA levels from WAT of the OXPHOS genes Cytochrome C (**d**), ATP synthase (**e**), Cytochrome C subunit 6 (**f**) and Cytochrome C subunit 5a (**g**) as well as mitochondrial regulators PGC-1a (**h**) and PGC-1b (**i**) and critical targets, such as ERRa (**j**) and PPARa (**k**). Error bars represent standard error. *t*-test was used for statistical analysis. The number of mice is indicated in each case. *$p < 0.05$. **$p < 0.01$. Source data are provided as a Source Data file

modified mouse models with delayed aging previously described. In particular, our model shows reduced body size, enhanced insulin sensitivity and even reduced incidence of fatal neoplastic disease, similar to the growth hormone (GH) mutant dwarf mice[57–60], although there are notable differences such as lower body fat, which dwarf mice tend to accumulate[60]. In this regard, mice with hyper-long telomeres are more similar to insulin-like growth factor (IGF) mutant mice, which also feature decreased body fat accumulation[60].

In summary, we demonstrate here that it is possible to generate mice that have telomeres which are much longer than those of the natural species by increasing the passage number of embryonic stem cells and in the absence of genetic modifications. These mice show a younger phenotype as indicated by improved mitochondrial function, improved metabolic parameters, decreased cancer, and increased longevity. These results also suggest that there is not a negative selection for individuals with longer telomeres than normal in species, and therefore, one can envision that natural selection processes which favor individuals with longer telomeres within a given species, could potentially increase species longevity. Finally, these findings also suggest the intriguing possibility that a potential mechanism for rending individuals with longer telomeres could be regulation of the duration of pluripotency stages during embryonic development when telomeres are elongated.

## Methods

**Chimaera generation**. ES cells were harvested from R1-eGFP 129S1 mouse embryos and cultured until passage 16 in 2i medium to allow telomere elongation[26]. Chimeric mice were generated at the CNIO Transgenic Mice Unit by microinjection of R1-eGFP ES cells into Hsd:ICR (C57Bl6 background). Embryos were harvested from super ovulated donor females at E2.5 days of gestation. Approximately 100 morulae at the eight-cell state were microinjected with 6–10 EGFP-expressing ES cells by laser-assisted perforation of the "zona pellucida". After microinjection, embryos were incubated overnight in a drop of KSOM medium in a $CO_2$ incubator at 37 °C under oil.

As a control for this study R1-eGFP 129S1 female mice were used in all the experiments.

**Mice**. All mice were maintained at the Spanish National Cancer Centre under specific pathogen-free conditions in accordance with the recommendations of the Federation of European Laboratory Animal Science Associations (FELASA). All animal experiments were approved by our Institutional Animal Care and Use Committee (IACUC) and by the Ethical Committee for animal experimentation (CEIyBA) (PROEX 133/15). We followed the Reporting in Vivo Experiments (ARRIVE) guidelines developed by the National Centre for the Replacement, Refinement & Reduction of Animals in Research (NC3Rs).

**Immunohistochemistry analyses in tissue sections**. Tissues were fixed in 10% buffered formalin, embedded in paraffin wax and sectioned at 5 mm. For histological examination sections were stained with hematoxylin and eosin, according to standard procedures. GFP (Cell Signaling, cat. number 2956) and p21 (291 H/B5, homemade) antibody was used for immunohistochemistry in tissue sections, both diluted 1:500. Nuclei were counterstained with Harrys's hematoxylin. Pictures were taken using Olympus AX70 microscope.

**Densitometry assay**. Body composition (percentage of total fat, bone densitometry and total lean) was measured using dual-energy X-ray absorptiometry (DEXA) PIXImus, Mouse Densitometer (GE Lunar co, Madison, WI, USA) using software version 1.46. Mice were anesthetized via gaseous anesthetic (isoflurane) with a continuous flow of 1% to 3% isoflurane/oxygen mixture (2 L/min), for ~10 min during recording. Quality control was performed using a calibrated phantom before imaging.

**Subcutaneous fat thickness**. Thickness of the subcutaneous fat layer was measured as previously described[61]. Briefly, a total of 30 measurements were performed on 2 back sections of the skin for each mouse around 100 weeks old. Measurements were made using Panoramic Viewer software and ImageJ software.

**Metabolic measurements**. Serum levels of albumin, creatinine, bilirubin, urea, alanine aminotransferase (ALT), cholesterol, low density lipoprotein (LDL) and high density lipoprotein (HDL) were determined using ABX Pentra (Horiba

Medical). To perform GTT and ITT, mice were i.p. injected with 2 g of glucose/kg of body weight and 0.75 U insulin/kg of body weight (Eli Lilly; Humalog Insulin), respectively. In the case of GTT mice were previously fasted for 16 h. For ITT mice were not fasted and basal levels of glucose were between 80–100 mg/dL.

**Cognitive tests**. To measure coordination mice were tested in a Rotarod apparatus (model LE 8200). For balance mice were tested by the tightrope test, as previously described[17,28]. To check smell capacities mice were tested by the buried food test[62]. Briefly, mice were fasted for 16 h and they were placed in a cage with a buried pellet of food. Analysis was made by calculating the percentage of success and the total time spent to find the pellet.

**Telomere length analyses**. Quantitative telomere fluorescence in situ hybridization (Q-FISH) was performed directly on paraffin-embedded lung sections as previously described[12] and analyzed by Definiens software. The incidence of short telomeres was calculated as the percentage of telomeres below the 25th percentile of telomere fluorescence in control samples.

**Real-time qPCR**. Total RNA from cells was extracted with the RNeasy kit (QIAGEN) and reverse transcribed was using the iSCRIPT cDNA synthesis kit (BIO-RAD) according to manufacturer's protocols. Quantitative real-time PCR was performed with the QuantStudio 6 Flex (Applied Biosystems, Life Technologies) using Go-Taq qPCR master mix (Promega) according to the manufacturer's protocol. Samples were run in triplicates. Primers were used as follows:

TBP1-F 5′-ACCCTTCACCAATGACTCCTATG-3′
TBP1-R 5′-TGACTGCAGCAAATCGCTTGG-3′
TRF1-F 5′-GTCTCTGTGCCGAGCCTTC-3′,
TRF1-R 5′-TCAATTGGTAAGCTGTAAGTCTGTG-3′
TRF2-F 5′-AGAGCCAGTGGAAAAACCAC-3′
TRF2-R 5′-ATGATGGGGATGCCAGATTA-3′
POT1A-F 5′-AAACTATGAAGCCCTCCCCA-3′
POT1A-R 5′-CGAAGCCAGAGCAGTTGATT-3′
POT1B-F 5′-AGTTATGGTCGTGGGATCAGAG -3′
POT1B-R 5′-GAGGTCTGAATGGCTTCCAA -3′
RAP1-F 5′-AAGGACCGCTACCTCAAGCA-3′
RAP1-R 5′-TGTTGTCTGCCTCTCCATTC-3′
TPP1-F 5′-ACTTGTGTCAGACGGAACCC-3′
TPP1-R 5′-CAACCAGTCACCTGTATCC-3′
TIN2-F 5′-TCGGTTGCTTTGCACCAGTAT-3′
TIN2-R 5′-GCTTAGCTTTAGGCAGAGGAC-3′
TERT-F 5′-GGATTGCCACTGGCTCCG-3′
TERT-R 5′-TGCCTGACCTCCTCTTGTGAC-3′
TERC-F 5′-TCATTAGCTGTGGGTTCTGGT-3′
TERC-R 5′-TGGAGCTCCTGCGCTGACGTT-3′
Cytochrome C-F 5′-ACCAAATCTCCACGGTCTGTT-3′
Cytochrome C-R 5′-GGATTCTCCAAATACTCCATCAG-3′
ATP synthase-F 5′-TCTCGGCCAGAGACTAGGAC-3′
ATP synthase-R 5′-GCACCTGCACCAATGAATTT-3′
Cytochrome c oxidase subunit 6-F 5′-GTAACGCTACTCCGGGACAA-3′
Cytochrome c oxidase subunit 6-R 5′-TCCAGGTAGTTCTGCCAACA-3′
Cytochrome c oxidase subunit 5a-F 5′-CTCGTCAGCCTCAGCCAGT-3′
Cytochrome c oxidase subunit 5a-R - 5′-TAGCAGCGAATGGAACAGAC-3′
PGC1α-F 5′-CCCTGCCATTGTTAAGACC-3′
PGC1α-R 5′-TGCTGCTGTTCCTGTTTTC-3′
PGC1β-F 5′-GGACGCCAGTGACTTTGACT-3′
PGC1β-R 5′-TTCATCCAGTTCTGGGAAGG-3′
ERRα-F 5′-CCTCTTGAAGAAGGCTTTGCA-3′
ERRα-R 5′-GCAGGGCAGTGGGAAGCTA-3′
PPARα-F 5′-TCGGCGAACTATTCGGCTG-3′
PPARα-F 5′-GCACTTGTGAAAACGGCAGT-3′

**Immunofluorescence analyses on tissue sections**. For immunofluorescence analyses, tissue sections were fixed in 10% buffered formalin (Sigma) and embedded in paraffin. After desparaffination and citrate antigen retrieval, sections were permeabilized with 0.5% Triton in PBS and blocked with 1% BSA and 10% Australian FBS (GENYCELL) in PBS. The antibodies were applied overnight in antibody diluents with background reducing agents (Invitrogen). Primary antibodies: anti-Rap1 (BL735, Bethyl, cat. number A300-306A), anti-53BP1 (Novus Biologicals, cat. Number NB100-304). Immunofluorescence images were obtained using a confocal ultraspectral microscope (Leica TCS-SP5). Quantifications were performed with Definiens software. A double immunofluorescence using antibodies against 53BP1 to mark DNA damage foci and TRF1 to mark telomeres was performed in tissue sections to assay for telomeric DNA damage specifically located at telomeres. All antibodies were diluted 1:500 for the experiments.

**Western-blot analyses**. Total protein extracts were obtained using RIPA extraction buffer and protein concentration was determined using the Bio-Rad DC Protein Assay (Bio-Rad). Up to 20 micrograms of protein per extract were separated in SDS–polyacrylamide gels by electrophoresis. After protein transfer onto

nitrocellulose membrane (Whatman), the membranes were incubated with the indicated antibodies. Antibody binding was detected after incubation with a secondary antibody coupled to horseradish peroxidase using chemiluminescence with ECL detection KIT (GE Healthcare). Primary antibodies: anti-TRF1 (BED5, Cell Signaling), anti-RAP1 (Bethyl), anti-SMC-1 (Bethyl). Quantifications: Protein-band intensities were measured with ImageJ software and normalized against the loading control.

**Mitochondrial copy number**. Relative mtDNA content were obtained by the comparative Ct method[63,64]. Briefly, we measured by qPCR the mtDNA genes *Cox1*, *Cytb* and *Nd1* and the nuclear DNA (nucDNA) gene *H19*. Then we subtracted the mtDNA averaged Ct values from the nucDNA averaged Ct values obtaining the ΔCt. We finally calculated the relative mitochondrial DNA content by raising 2 to the power of ΔCt and then multiplying by 2. Expressed as equations:

$$\Delta Ct = nucDNA\ Ct - mtDNA\ Ct.$$

Relative mitochondrial DNA content $= 2 \times 2^{\Delta Ct}$.

Primers were used as follows:

COX1-F 5′-CTGAGCGGGAATAGTGGGTA-3′
COX1-R 5′-TGGGGCTCCGATTATTAGTG-3′
CYTB-F 5′-ATTCCTTCATGTCGGACGAG-3′
CYTB-R 5′-ACTGAGAAGCCCCCTCAAAT-3′
ND1-F 5-′AATCGCCATAGCCTTCCTAACAT-3′
ND1-R 5′-GGCGTCTGCAAATGGTTGTAA-3′
H19-F 5′-GTACCCACCTGTCGTCC-3′
H19-R 5′-GTCCACGAGACCAATGACTG-3′

**Reporting summary**. Further information on research design is available in the Nature Research Reporting Summary linked to this article.

## Data availability

The data that support the findings of this study are available from the corresponding author upon reasonable request. The source data underlying Figs. 1–9 are provided as a Source Data file.

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

## Acknowledgements

We are indebted to D. Megias for microscopy analysis, to F. Mulero for molecular imaging, to R. Serrano for animal care, S. Ortega for chimaera generation and G. Bescós for helping with experiments. Research in the Blasco lab is funded by the Spanish Ministry of Economy and Competitiveness Projects (SAF2013-45111-R and SAF2015-72455-EXP), the Comunidad de Madrid Project (S2017/BMD-3770), the World Cancer Research (WCR) Project (16-1177), and the Fundación Botín (Spain).

## Author contribution

M.A.B had the original idea, secured funding, and supervised research. M.A.B. and M.A. M-L., wrote the paper. M.A.M.-L. performed experiments, analyzed the data. A.C.C.-M. performed experiments, including qPCR, IF and Q-FISH.

## Competing interests

The authors declare no competing interests.
