## [Peer Review File · Nature Communications]

Reviewers' comments:

Reviewer #1 (Remarks to the Author):

Muñoz-Lorente et al. make use of a previous described experimental model by the Blasco group. In previous work the group showed that ES cells derived from intracellular mass (ICM) of the early embryo exhibit progressive telomere elongation in culture compared to the early passage of freshly isolated ES cells. In the current study, the group investigated whether mice derived from early passage ES cells (normal telomeres) vs. mice derive from late passage ES cells (super long telomeres) would exhibit differences in lifespan and health parameters. To this end cohorts of aging mice from both groups were monitored during aging.

The study shows that mice with long telomeres exhibit a reduced body weight, an elongated lifespan, and decreased rates of spontaneous cancers compared to mice with normal telomere length. The authors show that the difference in body weight is due to a decrease in fat tissue in mice with long telomeres. This associates with reduced LDL levels and improved glucose responses in mice with hyper-long telomeres compared to normal telomere mice. The molecular analysis shows that increases in telomere length remain detectable in aging mice with hyper-long telomeres compared to mice with a normal telomere length. This associates with a reduction in telomere associated DNA damage foci (TIFs), DNA damage signaling (p21) and senescence in liver of aging mice with super long telomeres compared aging mice with normal telomere length. Of note, these findings do not associate with increases in the expression levels of component of the telomerase holoenzyme (Tert or Terc).

Critic: Overall this is an interesting study showing that elongated telomeres per se (without an increase in telomerase activity) have positive effects on health and lifespan by improving metabolic health in aging mice. While this observation is of general interest, I have a few major comments that should be addressed:

1. The authors should describe the generation of Morula complemented mice in greater detail. How exactly were the mice generated? How many cells were injected?
2. How did the authors ensure that during passage of the ICM-derived ES cell line, there was no genetic drift? Is it possible that subpopulations of more primitive ES cells grew out during passage? It could be useful to analyze the RNA expression profile of the ES cell lines that were used for generation of the mice, e.g. at early and late passage.
3. The authors used females ES cells that were injected into Morula to generate mice that were to 100% derived from the injected ES cells as demonstrated by the GFP marker expressed in the ES cells. The n-numbers in the study are at the lower end with only 10-16 mice per group for the survival studies. It is possible that the sex of the trophoblast tissue (which is embryo derived but not ES cell derived) has an impact on survival and there might be differences in the ratio of male to female embryo derived trophoblasts in the generated cohorts.
4. The authors should critically discuss these possible aspects in the revised discussion.
5. Mechanistically it remains unclear why mice with longer telomeres age better than mice with normal telomeres. The authors suggest that improvements in metabolic health parameters represent a causative factor for the improvement in health and lifespan extension in mice with hyper-long telomeres vs. controls. The metabolic parameters support this concept. However, the authors did not present any data on mitochondrial function, which would be important to directly support the concept that metabolic health is improved in mice with hyper-long telomeres. The authors should employ different assays to directly measure mitochondrial function in purified cell types of mice of the different cohorts, for example the seahorse assay could be employed.
6. Previous work on mice with short, dysfunctional telomeres revealed evidence that telomere dysfunction leads to suppression of mitochondrial biogenesis and function. These phenotypes were mediated in part by p53-activation and the application of diets with increases glucose content elongated the survival of telomere dysfunctional mice (see Sahin et al. 2011, Missios et al. 2014).

The current findings seem to stand in line with these previous findings, showing that hyper-long telomeres prevent activation of DNA damage markers and the coinciding development of metabolic compromise in aging mice. The authors should discuss the possible correlation of the current work with these previous studies on telomere dysfunction induced metabolic compromise.

Minor comments:

Is it possible that the data on 75 and 100 week old mice were mixed up in Fig. 3I-J?

The following statement in the discussion seems incomplete and lacks a reference: "In this regard, there is mounting evidence that telomerase re-activation in the adult organism by using non-integrative gene adeno associated vectors (AAV) does not lead to increased cancer, even in the context of activated oncogenes, in mice "

The authors state in the introduction and in the discussion that mouse telomere shorten 100 times faster than humans. Is there really good evidence for this? Isn't it possible that this reflects rather some kind of telomere breakage?

The method section is rather incomplete. I couldn't find any description on how the lean wait was determined.

Reviewer #2 (Remarks to the Author):

This manuscript describes impact of increased length of telomeres on multiple parameters related to metabolic regulation and aging. A unique type of mice with "hyper-long" telomeres was developed in the authors' laboratory and shown here to be healthy and long-lived with no indication of increased risk of cancer. Instead, the incidence of spontaneous tumors tends to be reduced (lack of evidence for significant reduction being likely related to the small number of animals). Physiological characteristics of these very interesting animals suggest slower rate of aging. Increased median and maximal longevity is consistent with this interpretation and invites comparisons with characteristics of animals with other anti-aging interventions. There are some striking similarities (e.g. enhanced insular sensitivity, reduced body size, reduced number of senescent cells) and also some differences (e.g. adiposity being reduced in some, but increased in other types of long-lived mice). Perhaps a few sentences could be added to the discussion to comment on these comparisons.

This exciting and well-written manuscript requires some fairly minor changes to clarify several issues and provide additional information on the methods. These and other issues are listed below.

p. 3, l. 22-24: This sentence needs to be rewritten

p. 4, l. 19-22 (and elsewhere in the manuscript): If these mice weighted less than normal controls and this was entirely due to reduced amount of WAT, then they would be more accurately described as "leaner," rather than "smaller." Was body length reduced? The figure suggests that I might have been. Reduced length would suggest a change in the growth rate of the skeleton (and justify the term "smaller). Please clarify!

p. 8, l. 12-19 and Methods: The graphs of GTT results indicate basal glucose levels of 30-40 mg/dL. This is very low. Presumably, the animals were fasted; for how long? What were basal glucose levels in ITT studies?

p. 9, l. 20-27: This apparent reduction would be more clearly described as 50% (approximately half of the increase seen in controls). The incidence of tumors in controls was 40%; do the authors have any information on the apparent causes of death in the remaining animals?

p. 10: It is arguable whether the employed tests should be referred to as tests of cognition. Speaking of endurance, coordination, and sensory perception would be more accurate.

p. 11, l. 14: "pronounced" or "consistent" rather than "significant"?

p. 11, l. 15/16: I believe this is the first mention of what the controls were. This should be stated earlier and more detail (breeding, etc.) provided in the Methods.

p. 13 l. 17: ...protein levels were...

p. 14, l. 5: ...shelterin components in...

p. 14, l. 4-6 & Fig 2E-2H: This is a minor point, but one could interpret these data as a very likely decrease with lack of significance due to having only five animals per group.

p. 15, l. 3: ...active in...

p. 15, l. 22: please add references

p. 15, l. 24: The phrase, "in large population studies," should be moved to the end of this sentence

p. 17: l. 22-24: This statement would benefit from a bit of expansion and clarification

Reviewed by,
Andrzej Bartke, Ph.D.

Detailed answers to reviewer #1:

[REVIEWER] Muñoz-Lorente et al. make use of a previous described experimental model by the Blasco group. In previous work the group showed that ES cells derived from intracellular mass (ICM) of the early embryo exhibit progressive telomere elongation in culture compared to the early passage of freshly isolated ES cells. In the current study, the group investigated whether mice derived from early passage ES cells (normal telomeres) vs. mice derive from late passage ES cells (super long telomeres) would exhibit differences in lifespan and health parameters. To this end cohorts of aging mice from both groups were monitored during aging.

The study shows that mice with long telomeres exhibit a reduced body weight, an elongated lifespan, and decreased rates of spontaneous cancers compared to mice with normal telomere length. The authors show that the difference in body weight is due to a decrease in fat tissue in mice with long telomeres. This associates with reduced LDL levels and improved glucose responses in mice with hyper-long telomeres compared to normal telomere mice. The molecular analysis shows that increases in telomere length remain detectable in aging mice with hyper-long telomeres compared to mice with a normal telomere length. This associates with a reduction in telomere associated DNA damage foci (TIFs), DNA damage signaling (p21) and senescence in liver of aging mice with super long telomeres compared aging mice with normal telomere length. Of note, these findings do not associate with increases in the expression levels of component of the telomerase holoenzyme (Tert or Terc).

Critic: Overall this is an interesting study showing that elongated telomeres per se (without an increase in telomerase activity) have positive effects on health and lifespan by improving metabolic health in aging mice.

[ANSWER] We deeply appreciate the commentaries by this reviewer and are very glad that the reviewer finds the study interesting.

This reviewer has the following major issues with this manuscript:

[REVIEWER] The authors should describe the generation of Morula complemented mice in greater detail. How exactly were the mice generated? How many cells were injected?

[ANSWER] Although the details of mice generation and morula microinjection were already explained in the "Material and Methods" section of the manuscript (see page 21, lines 4-12), in the revised version of the manuscript, we also include this information in the "Results" section (see page 6, lines 9-13).

[REVIEWER] How did the authors ensure that during passage of the ICM-derived ES cell line, there was no genetic drift? Is it possible that subpopulations of more primitive ES cells grew out during passage? It could be useful to analyze the RNA expression profile of the ES cell lines that were used for generation of the mice, e.g. at early and late passage.

[ANSWER] The reviewer has a very good point. We already addressed this important point in a previous paper from our lab (Varela et al., *Nat Comm.* 2016) by performing RNA sequencing of early and late passage ES cells, as suggested by the reviewer (see Supplementary Table 1 in Varela et al., *Nat Comm.* 2016). In particular, we performed RNA sequencing of four independent ES cells clones at early and late passage. We only found five genes out of 19,555 analyzed genes that were differentially expressed. These genes were Sox18 (upregulated 20-fold compared to ES cells at early passage and Sox17, Zbtb48, Chst15 and Jph4 (downregulated less than 2-fold compared to ES cells at early passage). Thus, the analysis showed very few changes in gene expression between the two groups. We now mention this in the revised manuscript text (see page 4, lines 16-17).

[REVIEWER] The authors used females ES cells that were injected into Morula to generate mice that were to 100% derived from the injected ES cells as demonstrated by the GFP marker expressed in the ES cells. The n-numbers in the study are at the lower end with only 10-16 mice per group for the survival studies. It is possible that the sex of the trophoblast tissue (which is embryo derived but not ES cell derived) has an impact on survival and there might be differences in the ratio of male to female embryo derived trophoblasts in the generated cohorts.

[ANSWER] This is an interesting comment by the reviewer. It is true that trophoblast tissue is derived from the receptor morula and not from the ES cells microinjected. As we describe in “Material and Methods” and “Results” section of the manuscript, we microinjected around 100 Morulae for chimaera generation. Since the mice obtained were 100% derived from our GFP ES cells it was not possible to determine the original gender of the donor morula in adult mice. However, it is known (Baines and Renaud, 2017; Prudhomme and Morey, 2016) that trophoblast cells only contribute to placenta and amniotic sac formation, suggesting that gender of those cells are not affecting adult mice in any ways since nothing of them remains in them.

[REVIEWER] The authors should critically discuss these possible aspects in the revised discussion.

[ANSWER] As suggested by the reviewer, we now include a sentence discussing these issues (see page 19, lines 13-18).

[REVIEWER] Mechanistically it remains unclear why mice with longer telomeres age better than mice with normal telomeres. The authors suggest that improvements in metabolic health parameters represent a causative factor for the improvement in health and lifespan extension in mice with hyper-long telomeres vs. controls. The metabolic parameters support this concept. However, the authors did not present any data on mitochondrial function, which would be important to directly support the concept that metabolic health is improved in mice with hyper-long telomeres. The authors should employ different assays to directly measure mitochondrial function in purified cell types of mice of the different cohorts, for example the seahorse assay could be employed.

[ANSWER] This is a very interesting suggestion by the reviewer. To address mitochondrial function, we have performed a PCR based assay to determine relative mtDNA copy number in the white adipose tissue (WAT) of these mice using three different mitochondrial genes (*Cox1*, *Cytb* and *Nd1*). We found that hyper-long telomere mice show increased relative mtDNA copy number compared to normal length controls (see **new Figure 7A-C**). Moreover, we also measured mRNA expression levels of the different oxidative phosphorylation genes (OXPHOS) Cytochrome C, ATP synthase, Cytochrome C subunit 6 and Cytochrome C subunit 5a as well as mitochondrial regulators PGC-1 α and PGC-1 β and critical targets such as ERR α and PPAR α . Again, we found a significant upregulation in all these genes in hyper-long telomere mice compared to age-matched controls (see **new Figure 7D-K**).

[REVIEWER] Previous work on mice with short, dysfunctional telomeres revealed evidence that telomere dysfunction leads to suppression of mitochondrial biogenesis and function. These phenotypes were mediated in part by p53-activation and the application of diets with increases glucose content elongated the survival of telomere dysfunctional mice (see Sahin et al. 2011, Missios et al. 2014). The current findings seem to stand in line with these previous findings, showing that hyper-long telomeres prevent activation of DNA damage markers and the coinciding development of metabolic compromise in aging mice. The authors should discuss the possible correlation of the current work with these previous studies on telomere dysfunction induced metabolic compromise.

[ANSWER] We thank the reviewer for this point of view and we have now included this in the revised version of the manuscript (see page 18, lines 19-24 and page 19, lines 1-6).

Minor comments with this manuscript:

[REVIEWER] Is it possible that the data on 75 and 100 week old mice were mixed up in Fig. 3I-J?

[ANSWER] We did not find any error in the mentioned figures.

[REVIEWER] The following statement in the discussion seems incomplete and lacks a reference: "In this regard, there is mounting evidence that telomerase re-activation in the adult organism by using non-integrative gene adeno associated vectors (AAV) does not lead to increased cancer, even in the context of activated oncogenes, in mice"

[ANSWER] We have now included the missing references in the revised manuscript text (see page 17, lines 2-3).

[REVIEWER] The authors state in the introduction and in the discussion that mouse telomere shorten 100 times faster than humans. Is there really good evidence for this? Isn't it possible that this reflects rather some kind of telomere breakage?

[ANSWER] There is previous evidence that mice telomeres shorten faster than human telomeres in longitudinal studies (Vera et al., 2012). We cite this paper in the manuscript text.

[REVIEWER] The method section is rather incomplete. I couldn't find any description on how the lean wait was determined.

[ANSWER] We have now extended this information in the revised manuscript text (see page 22, lines 13-14).

Detailed answers to reviewer #2:

This reviewer has the following major issues with this manuscript:

[REVIEWER] This manuscript describes impact of increased length of telomeres on multiple parameters related to metabolic regulation and aging. A unique type of mice with “hyper-long” telomeres was developed in the authors’ laboratory and shown here to be healthy and long-lived with no indication of increased risk of cancer. Instead, the incidence of spontaneous tumors tends to be reduced (lack of evidence for significant reduction being likely related to the small number of animals). Physiological characteristics of these very interesting animals suggest slower rate of aging. Increased median and maximal longevity is consistent with this interpretation and invites comparisons with characteristics of animals with other anti-aging interventions. There are some striking similarities (e.g. enhanced insular sensitivity, reduced body size, reduced number of senescent cells) and also some differences (e.g. adiposity being reduced in some, but increased in other types of long-lived mice). Perhaps a few sentences could be added to the discussion to comment on these comparisons.

[ANSWER] We thank the reviewer for all the insightful commentaries. In the revised manuscript text, we now have extended the discussion by including these ideas (see page 19, lines 19-25 and page 20, lines 1-4).

[REVIEWER] This exciting and well-written manuscript requires some fairly minor changes to clarify several issues and provide additional information on the methods. These and other issues are listed below.

[ANSWER] We sincerely thank the reviewer for considering that this is an exciting and well-written manuscript. In the revised version of the manuscript we have clarified the issues raised by the reviewer and provide the requested information.

Minor comments with this manuscript:

[REVIEWER] p. 3, l. 22-24: This sentence needs to be rewritten

[ANSWER] we have change the sentence in the revised manuscript (see page 4, lines 3-6).

[REVIEWER] p. 4, l. 19-22 (and elsewhere in the manuscript): If these mice weighted less than normal controls and this was entirely due to reduced amount of WAT, then they would be more accurately described as “leaner,” rather than “smaller.” Was body length reduced? The figure suggests that I might have been. Reduced length would suggest a change in the growth rate of the skeleton (and justify the term “smaller). Please clarify!

[ANSWER] The hyper-long telomere mice have reduced body fat. We have now referred to “reduced body size” before determination of body fat, and “leaner” once we determined the cause of the reduced body size (see page 2, line 12; page 5, line 4; page 6, line 22; page 7, line 5; page 18 line 10; and page 39, line 3).

[REVIEWER] p. 8, l. 12-19 and Methods: The graphs of GTT results indicate basal glucose levels of 30-40 mg/dL. This is very low. Presumably, the animals were fasted; for how long? What were basal glucose levels in ITT studies?

[ANSWER] We agree with the reviewer in this point. Mice were fasted in GTT experiments for 16h, which explain the low levels of basal glucose. For ITT experiment basal glucose levels were between 80-100 mg/dL since they were not fasted. In the revised manuscript, we now include fasting periods and basal glucose levels in both sections in the revised manuscript (see page 8, lines 18,19 and page 23, lines 13,14).

[REVIEWER] p. 9, l. 20-27: This apparent reduction would be more clearly described as 50% (approximately half of the increase seen in controls). The incidence of tumors in controls was 40%; do the authors have any information on the apparent causes of death in the remaining animals?

[ANSWER] We agree with the reviewer and we have modified the text accordingly (see page 10, lines 2-4). Regarding the cause of death, was mostly caused by general body degeneration due to the aging process, and some of them presented infected uterus, a common pathology in aged female mice. We now include this information in the revised manuscript text (see page 10, lines 4-7).

[REVIEWER] p. 10: It is arguable whether the employed tests should be referred to as tests of cognition. Speaking of endurance, coordination, and sensory perception would be more accurate.

[ANSWER] We fully agree with the reviewer and we have changed the wording in the revised manuscript (see page 10, lines 16, 20 and 24).

[REVIEWER] p. 11, l. 14: “pronounced” or “consistent” rather than “significant”?

[ANSWER] We have changed this in the revised manuscript (see page 11, line 22).

[REVIEWER] p. 11, l 15/16: I believe this is the first mention of what the controls were. This should be stated earlier and more detail (breeding, etc.) provided in the Methods.

[ANSWER] We have now included a more detailed information in the revised manuscript text (see page 21, lines 13, 14).

[REVIEWER] p. 13 l. 17: ...protein levels were...

[ANSWER] We have now changed this typo in the revised manuscript text.

[REVIEWER] p. 14, l. 5: ...shelterin components in...

[ANSWER] We have now changed this typo in the revised manuscript text.

[REVIEWER] p. 14, l. 4-6 & Fig 2E-2H: This is a minor point, but one could interpret

these data as a very likely decrease with lack of significance due to having only five animals per group.

[ANSWER] We appreciate the consideration of the reviewer in this point and we have toned this down in the revised manuscript text (page 15, lines 9-12).

[REVIEWER] p. 15, l. 3: ...active in...

[ANSWER] We will change this typo in the revised manuscript.

[REVIEWER] p. 15, l. 22: please add references

[ANSWER] We have now added the missing references in the revised manuscript text (see page 17, lines 2-3).

[REVIEWER] p. 15, l. 24: The phrase, "in large population studies," should be moved to the end of this sentence

[ANSWER] We have now moved sentence as the reviewer suggests in the revised manuscript (see page 17, lines 4-5).

[REVIEWER] p. 17: l 22-24: This statement would benefit from a bit of expansion and clarification.

[ANSWER] We have now expanded this sentence (page 20, lines 6-9).

References:

Baines, K.J., and Renaud, S.J. (2017). Transcription Factors That Regulate Trophoblast Development and Function. In Progress in Molecular Biology and Translational Science, p.

Prudhomme, J., and Morey, C. (2016). Epigenesis and plasticity of mouse trophoblast stem cells. Cell. Mol. Life Sci.

Varela, E., Muñoz-Lorente, M.A., Tejera, A.M., Ortega, S., and Blasco, M.A. (2016). Generation of mice with longer and better preserved telomeres in the absence of genetic manipulations. Nat. Commun. 7.

REVIEWERS' COMMENTS:

Reviewer #1 (Remarks to the Author):

The authors nicely addressed my comments. A few minor points should be addressed:

1. Following sentence in the result section should be modified:

"Strikingly, these mice 100% contributed by ES cells with hyper-long telomeres showed a reduced body size compared to control mice from the same genetic background (Fig. 1C)."

The sentence is difficult to understand. It may be better to split it into two sentences. Also the comment of reviewer 2 was that these mice are leaner, not smaller. I agree with this comment. If the mice have less WAT but are not smaller in terms of body length, then it is better to describe the mice as "leaner" not "smaller".

Also in the following paragraphs the authors describe the mice as small:

"Thus, here we generated viable mice that are 100% contributed by ES cells with hyper-long telomeres. We did not find any overt phenotypes in these mice other than they have a reduced body size than that of the normal species.

Less fat accumulation in hyper-long telomere mice
In order to investigate the small-body phenotype...."

I think the authors should replace "small" by "lean" throughout the manuscript. It is surprising that in response to reviewer-2, the authors indicated that they would already have conducted this change in the revised manuscript, which apparently is not the case?!

The authors reference two papers from the DePinho lab to discuss previous findings that have shown that telomere dysfunction can lead to impaired mitochondrial function. One of the two papers is a review. It would be better to cite the second original paper, which showed telomere dysfunctional mice have impaired mitochondrial biogenesis and function as indicated in my first review:

Sahin et al. Nature 2011
Missios et al. Nat Comm. 2014

Same holds true for the last part of the discussion. In both paragraphs the review from 2012 could be deleted if the number of references is an issue.

Reviewer #2 (Remarks to the Author):

The authors dealt thoroughly with all of my comments. I believe they also dealt well with the comments of Reviewer 1 (although she/he can, of course, judge this better) and that this important paper was improved as a result. One minor point, line 299 (page 14)- ...were double those of....

Andrzej Bartke

Detailed answers to reviewer #1:

[REVIEWER] 1. Following sentence in the result section should be modified: "Strikingly, these mice 100% contributed by ES cells with hyper-long telomeres showed a reduced body size compared to control mice from the same genetic background (Fig. 1C)." The sentence is difficult to understand. It may be better to split it in two sentences. Also the comment of reviewer 2 was that these mice are leaner, not smaller. I agree with this comment. If the mice have less WAT but are not smaller in terms of body length, then it is better to describe the mice as "leaner" not "smaller".

[ANSWER] We agree with the reviewer in this statement and we have changed the sentence to be more clear and properly described mice as leaner instead of smaller (see page 6, lines 20-22).

[REVIEWER] Also in the following paragraphs the authors describe the mice as small: "Thus, here we generated viable mice that are 100% contributed by ES cells with hyper-long telomeres. We did not find any overt phenotypes in these mice other than they have a reduced body size than that of the normal species.

Less fat accumulation in hyper-long telomere mice

In order to investigate the small-body phenotype....."

I think the authors should replace "small" by "lean" throughout the manuscript. It is surprising that in response to reviewer-2, the authors indicated that they would already have conducted this change in the revised manuscript, which apparently is not the case?!

[ANSWER] As the reviewer suggested we have replaced small by lean in the revised manuscript (see page 7, line 3 and 6).

[REVIEWER] The authors reference two papers from the DePinho lab to discuss previous findings that have shown that telomere dysfunction can lead to impaired mitochondrial function. One of the two papers is a review. It would be better to cite the second original paper, which showed telomere dysfunctional mice have impaired mitochondrial biogenesis and function as indicated in my first review:

Sahin et al. Nature 2011

Missios et al. Nat Comm. 2014

Same holds true for the last part of the discussion. In both paragraphs the review from 2012 could be deleted if the number of references is an issue.

[ANSWER] We have changed the mentioned reference in both parts of the manuscript (see page 13 line 9 and page 18 line 11).

Detailed answers to reviewer #2:

[REVIEWER] The authors dealt thoroughly with all of my comments. I believe they also dealt well with the comments of Reviewer 1 (although she/he can, of course, judge this better) and that this important paper was improved as a result. One minor point, line 299 (page 14)- ...were double those of....

[ANSWER] We thank the reviewer for all the insightful commentaries. We have modified the sentence in the revised manuscript (see page 13 lines 14 and 15).